# Expression and Interaction Proteomics of GluA1- and GluA3-Subunit-Containing AMPARs Reveal Distinct Protein Composition

**DOI:** 10.3390/cells11223648

**Published:** 2022-11-17

**Authors:** Sophie J. F. van der Spek, Nikhil J. Pandya, Frank Koopmans, Iryna Paliukhovich, Roel C. van der Schors, Mylene Otten, August B. Smit, Ka Wan Li

**Affiliations:** Department of Molecular and Cellular Neurobiology, Center for Neurogenomics and Cognitive Research, Amsterdam Neuroscience, Vrije Universiteit Amsterdam, 1081 HV Amsterdam, The Netherlands

**Keywords:** quantitative proteomics, synapse, protein complex, AMPA receptor

## Abstract

The AMPA glutamate receptor (AMPAR) is the major type of synaptic excitatory ionotropic receptor in the brain. AMPARs have four different subunits, GluA1–4 (each encoded by different genes, *Gria1*, *Gria2*, *Gria3* and *Gria4*), that can form distinct tetrameric assemblies. The most abundant AMPAR subtypes in the hippocampus are GluA1/2 and GluA2/3 heterotetramers. Each subtype contributes differentially to mechanisms of synaptic plasticity, which may be in part caused by how these receptors are regulated by specific associated proteins. A broad range of AMPAR interacting proteins have been identified, including the well-studied transmembrane AMPA receptor regulatory proteins TARP-γ2 (also known as Stargazin) and TARP-γ8, Cornichon homolog 2 (CNIH-2) and many others. Several interactors were shown to affect biogenesis, AMPAR trafficking, and channel properties, alone or in distinct assemblies, and several revealed preferred binding to specific AMPAR subunits. To date, a systematic specific interactome analysis of the major GluA1/2 and GluA2/3 AMPAR subtypes separately is lacking. To reveal interactors belonging to specific AMPAR subcomplexes, we performed both expression and interaction proteomics on hippocampi of wildtype and *Gria1*- or *Gria3* knock-out mice. Whereas GluA1/2 receptors co-purified TARP-γ8, synapse differentiation-induced protein 4 (SynDIG4, also known as Prrt1) and CNIH-2 with highest abundances, GluA2/3 receptors revealed strongest co-purification of CNIH-2, TARP-γ2, and Noelin1 (or Olfactomedin-1). Further analysis revealed that TARP-γ8-SynDIG4 interact directly and co-assemble into an AMPAR subcomplex especially at synaptic sites. Together, these data provide a framework for further functional analysis into AMPAR subtype specific pathways in health and disease.

## 1. Introduction

AMPA receptors (AMPARs) are glutamate-gated cationic channels underlying the predominant component of fast excitatory synaptic transmission in the mammalian central nervous system. Functional synaptic AMPARs are localized primarily in nano-domains [1,2] residing at the postsynaptic density, in which they are aligned to the glutamate-release sites of the presynaptic active zone [3]. AMPAR subunit composition, phosphorylation state, numbers and biophysical properties are regulated in an activity-dependent manner, which is a major postsynaptic contribution to alteration of synaptic efficacy. AMPARs have four different subunits GluA1–4 (each encoded by different genes, *Gria1*, *Gria2*, *Gria3* and *Gria4*), that can form distinct combinations in a tetrameric assembly [4,5]. In hippocampus, the majority of AMPARs consists of GluA1/2 followed by GluA2/3 heterotetramers [6,7]. The distinct AMPAR subunits show differences in levels of expression [7,8,9], posttranslational modifications [10], subcellular distribution [8,11], trafficking behavior [12,13] and channel properties [11,14], and contribute differentially to mechanisms of synaptic plasticity [13,14]. For instance, GluA1/2 receptors are inserted into the synapse upon stimulation, whereas GluA2/3 receptor cycle constitutively under basal conditions [13]. Interestingly, previous work revealed that endocytosis of GluA3-containing AMPARs specifically is required for the effects of Amyloid-β, one of the hallmarks of Alzheimer’s disease [15]. Unraveling the distinct regulation of AMPAR subtypes is therefore important to understand synaptic functioning in both health and disease.

These specific properties of AMPARs are generated by interactions with associated (auxiliary) proteins [10,16,17]. As such these proteins may cause different subunits to become differentially implicated in distinct phases of memory [18] and disease [15].

Interaction proteomics has been used to identify AMPAR associated proteins [17,19]. These receptor interactors include multiple membrane proteins, that are considered auxiliary proteins due to their effects on both AMPAR gating properties as well as trafficking [17,20]. In particular, TARP-γ2 (or Stargazin) and TARP-γ8 are known to alter AMPAR surface expression [21]; to affect AMPAR post-synaptic density (PSD) mobility by the interaction with PSD-95 [22,23]; and to prolong AMPAR deactivation and desensitization [24]. Apart from TARPs, CNIH-2/3 can regulate AMPAR channel properties [20,25]. The Shisa family member proteins Shisa6 and Shisa9 (also known as CKAMP52 and CKAMP44, respectively) have also been implicated in affecting AMPAR membrane mobility [26] and channel conductance properties [16,26,27,28].

Previous studies revealed several interactors gathering in distinct AMPAR assemblies [29,30]. For instance, FRRS1L together with CPT1c is located in the ER forming a subcomplex that regulates AMPAR biogenesis [29,30]. This complex is distinct from synaptic complexes containing, for example, the high abundant AMPAR interactor TARP-γ8 [29,30]. Both TARP-γ8 and FRRS1L compete for the same binding site on the AMPAR [31], and are therefore part of at least two separate AMPAR populations [29]. In addition, several AMPAR interactors revealed preferred association with distinct AMPAR subunits [18,32]. For instance, the classical AMPAR interactor SAP97 (or Dlg1) specifically binds GluA1 [33,34]. Additionally, unlike GluA1, the GluA2 and GluA3 subunits contain a shared sequence (-SKVI) at their C-terminal end. Through this sequence GRIP-1 [35] and PICK1 [36] interact with the AMPAR and regulate insertion and retainment [37,38] and removal [39] of the AMPAR from the synapse, respectively.

The two major AMPAR subtypes GluA1/2 and GluA2/3 in hippocampus contribute differentially to synaptic plasticity [13,14], and disease [15] which may in part be brought about by that these receptors are regulated by distinct interactors. Several studies demonstrated that certain interacting proteins associate differentially to specific AMPAR subunits [16,33,34,40]. However, an interactome analysis of the GluA1/2 and GluA2/3 subtypes in isolation is lacking. In the current study, we set out to determine the GluA1/2 and GluA2/3 complex compositions separately using expression and interaction proteomics, and super-resolution microscopy. We revealed strong co-occurrence of TARP-γ8, SynDIG4 (also known as Prrt1) and CNIH-2 with the GluA1/2 receptor specifically. In contrast, GluA2/3 revealed the most abundant association with TARP-γ2, CNIH-2 and Noelin1 (or Olfactomedin-1). Further analysis revealed a direct interaction between TARP-γ8 and SynDIG4, and their co-assembly into an AMPAR subcomplex, especially near the synapse. Together, these data provide a framework for further functional analysis into AMPAR subtype specific behaviors.

## 2. Materials and Methods

### 2.1. Animals

*Gria1*- and *Gria3* knock-out (KO) mice were obtained from the Gria1tm3Rlh/J [41] and Gria3tm1Dgen/Mmnc (RRID: MMRRC_030969-UNC) (MMRRC, Davis, CA, USA) mouse lines, respectively, crossed with C57BL6. All breedings were approved by The Netherlands central committee for animal experiments (CCD) and the animal ethical care committee (DEC) of the Vrije Universiteit Amsterdam.

Of note, *Gria1* KO mice revealed strong reduction in the expression of α-synuclein. Loss of α-synuclein expression has been observed previously in a sub population of C57BL/6J mice without alteration of additional genes or a noticeable phenotype [42]. As reduced α-synuclein in the current study is likely due to cross breeding with this C57BL/6J strain, we removed this protein from further analysis.

### 2.2. Antibodies

Detailed information on the antibodies used is shown in the Appendix A.

### 2.3. Preparation of Crude Synaptosomal Fractions

Biochemical fractions containing crude synaptosomes and microsomes (P2+M) were prepared as previously described [43] (Appendix A).

### 2.4. Immuno-Purifications/in-Gel Digestion/Data-Dependent Acquisition Analysis

Proteins were extracted from P2+M using n-Dodecyl β-D-maltoside (DDM) (Thermo Fisher, Waltham, MA, USA) dissolved in sample suspension buffer (25 mM, 150 mM NaCl and protease inhibitor cocktail (Roche, Basel, Switzerland), pH 7.4), at a 1% end-concentration, two times for 1 h at 4 °C. Following each extraction, samples were centrifuged at 20,000× *g* for 20 min. Next, supernatant was incubated with 10 µg of antibody overnight at 4 °C, followed by incubation with 80 µL of protein A/G PLUS-Agarose beads (Santacruz, Dallas, TX, USA) for 1 h at 4 °C. Samples were centrifuged at 1000× *g* for 1 min, supernatant was discarded and beads were washed four times with 1 mL washing buffer containing 0.1% DDM, 150 mM NaCl (Sigma-Aldrich, St. Louis, MO, USA), 250 mM HEPES (Sigma-Aldrich, St. Louis, MO, USA), pH 7.4. SDS sample buffer was added to the final pellet, samples were heated at 98 °C and run on a home-made 10% SDS polyacrylamide gel. All reported n-numbers are biological replicates.

Gels were fixed overnight in 50% ethanol and 3% phosphoric acid (Sigma-Aldrich, St. Louis, MO, USA), washed in MilliQ water and stained with Colloidal Coomassie Blue. Each sample lane was cut in 3–5 slices that were subsequently cut into smaller pieces. The gel pieces were transferred to a Multiscreen HV filter Plate (Sigma-Aldrich, St. Louis, MO, USA), washed and destained with a mixture of 50 mM ammonium bicarbonate (Sigma-Aldrich, St. Louis, MO, USA) in acetonitrile (VWR, Radnor, PA, USA). The gel pieces were dried with 100% acetonitrile and incubated overnight at 37 °C with trypsin (Mass Spec Grade, Promega, Madison, WI, USA) dissolved in 50 mM ammoniumbicarbonate. Peptides were extracted twice in 0.1% Trifluoroacetic acid (Protein sequence grade; Applied Biosystems, Warrington, UK) and 50% acetonitrile, followed by extraction in 0.1% Trifluoroacetic acid and 80% acetonitrile. Subsequently the samples were dried in a speed vac (Savant, Thermo Fisher, Waltham, MA, USA) and stored at −20 °C until mass spectrometry analysis. 

Peptides were analyzed on an LTQ-Orbitrap discovery (Thermo Fisher, Waltham, MA, USA) mass spectrometer as previously described [43], with some modifications (Appendix A).

### 2.5. Depletion Immuno-Purifications

Depletion Immuno-purifications (IPs) were performed using a similar protocol as described for the regular IPs, with some modifications (Appendix A).

### 2.6. Immuno-Purifications/Blue Native-PAGE/Data-Dependent Acquisition Analysis

IPs were performed using the protocol described above, now using 30 mg P2+M, 100 μL antibody and 1000 μL of beads. After purification and washing of the samples, purified protein complexes were eluted twice using 500 µg peptide dissolved in 1 mL washing buffer for 1 h. The samples were then concentrated using a 30 kDa filter (Bio-Rad, Hercules, CA, USA) for 30 min, and mixed with Blue Native (BN)-PAGE loading buffer (Thermo Fisher, Waltham, MA, USA), 0.5 µL molecular weight marker (Thermo Fisher, Waltham, MA, USA), 1 µL Coomassie G-250 mix (Thermo Fisher, Waltham, MA, USA). Samples were run on a 3–12% polyacrylamide precast BN-PAGE gel (Thermo Fisher, Waltham, MA, USA), at 1 mA constant current for 1 h and 2 mA constant current for 16 h at 4 °C.

Gels were fixed overnight in 50% ethanol, 3% phosphoric acid, washed in MilliQ water and stained with Colloidal Coomassie Blue. Each sample was cut into 70 slices using a grid cutter (Gel Company, San Francisco, CA, USA), and transferred to a Multiscreen HV filter Plate. Cysteines were derivatized using 1 mM tris(2-carboxyethyl)phosphine (TCEP) (Sigma-Aldrich, St. Louis, MO, USA) in 50 mM ammonium bicarbonate for 30 min at 37 °C and incubated with 4 mM methyl methanethiosulfonate (MMTS) (Fluka, Honeywell, Charlotte, NC, USA) in 50 mM ammonium bicarbonate for 15 min at room temperature. Next, samples were washed, destained, dried and digested following the in-gel digestion protocol described above. The samples were dried in a speed vac and stored at −20 °C before analysis on the mass spectrometer.

Each slice was analyzed separately on the Triple TOF 5600 (Sciex, Framingham, MA, USA) in data-dependent acquisition (DDA) mode as described previously [44], with some modifications (Appendix A).

### 2.7. Co-Purification from HEK293 Cells

HEK293 cells were plated in 10 cm dishes in Dulbecco’s modified Eagle’s medium (DMEM, Gibco, Life Technologies, Carlsbad, CA, USA) supplemented with 10% fetal bovine serum (Invitrogen, Waltham, MA, USA) and 1% penicillin-streptomycin (Gibco, Life Technologies, Carlsbad, CA, USA) and kept at 37 °C, 95% air and 5% CO_2_. At ~70% confluency, cells were transfected using polyethylenimine (PEI) and 5 μg plasmid cDNA for TARP-γ8-Myc and SynDIG4-HA.

After 48 h, the HEK293 cells were washed with phosphate-buffered saline resuspended in extraction buffer (1% DDM, 25 mM HEPES, 150 mM NaCl, and protease inhibitor cocktail, pH 7.4), and incubated for 1 h at 4 °C. After two consecutive centrifugation steps at 20,000× *g* for 15 min. 4 °C, 4 ug of antibody was added to the supernatant, incubated overnight at 4 °C, followed by 1 h incubation with beads, 4 °C. The samples were washed four times with wash buffer (0.1% DDM, 25 mM HEPES, and 150 mM NaCl) in between centrifugation at 1000× *g*, 4 °C, and the purified proteins were eluted—with 2× sodium dodecyl sulfate (SDS) sample buffer. Input samples were prepared from the supernatant fraction by addition of SDS sample buffer to a 2× final concentration.

### 2.8. BN-PAGE/Immunoblot Analysis

BN-PAGE for immunoblot analysis was performed following the manufacturer’s recommendations (Thermo Fisher, Waltham, MA, USA), with some modification (Appendix A). Immunoblot analysis was conducted following the regular immunoblot protocol described in the Appendix A.

### 2.9. Quantitative Proteomics by in-Gel Digestion/Data-Independent Acquisition

Wildtype, *Gria1*- and *Gria3* KO P2+M samples were run on a home-made 10% SDS polyacrylamide gel. Each sample was cut into small pieces, 100 μL of 50 mM ammoniumbicarbonate and 5 mM TCEP was added and incubated for 30 min, 37 °C. Next, 100 μL of 50 mM ammoniumbicarbonate and 2.5 mM MMTS was incubated for 15 min, room temperature. The proteins were digested using the in-gel digestion protocol described above. All reported n-numbers are biological replicates.

Peptides were analyzed by micro-Liquid Chromatography–Tandem Mass Spectrometry (LC-MS/MS) using an Ultimate 3000 LC system (Dionex, Thermo Fisher, Waltham, MA, USA) coupled to the TripleTOF 5600 mass spectrometer (Sciex, Framingham, MA, USA). Analysis was performed in Data-Independent Acquisition (DIA also known as SWATH) mode, as described previously [45,46,47], with some modifications (Appendix A).

### 2.10. Primary Neuronal Culture

Detailed information on the preparation of dissociated hippocampal neuronal cultures is shown in Appendix A.

### 2.11. Immunocytochemistry

Detailed information on immunolabeling of hippocampal neurons is shown in Appendix A.

### 2.12. STED Microscopy and Analysis

Images were acquired on a TCS SP8 gated Stimulated Emission Depletion (STED) 3X Microscope (Leica, Wetzlar, Germany). Fluorophores were excited with a pulsed white light laser at their excitation peak, and a pulsed 775 nm STED laser was used for depletion in the 635 nm (TARP-γ8) and 580 nm (SynDIG4) channel obtaining a lateral resolution of ~80 nm. Images in the 488 nm (Homer) channel were taken in confocal mode. Images were obtained with a 100× oil objective (NA = 1.4), a mechanical zoom of 5 and the pinhole set at 1 Airy Units (AU). Signals were detected with a gated hybrid detector (HyD) set in photon counting mode.

The Images were deconvolved with Huygens Software (Scientific Volume Imaging B.V., Hilversum, The Netherlands) using the Good’s Roughness Maximum Likelihood Estimation (GMLE) algorithm and analyzed with ImageJ extended in the Fiji framework. Analysis was performed on the maximum projections of the z-stack, and a threshold determined by the default algorithm was applied on all channels. The Manders’ coefficients were obtained in the coloc2 application.

## 3. Results

### 3.1. Expression Proteomics on Gria1- and Gria3 KO Synapses Reveals Differential Expression of Known AMPAR Interactors

We first performed quantitative proteomics on hippocampal synapse enriched fractions of both *Gria1*- and *Gria3* KO mice and their wildtype controls (n = 5–6/condition) (Figure 1). Per dataset, differential expression analysis (DEA) was performed using high-quality peptides detected in at least 75% of the samples in each experimental condition. In addition, ambiguous peptides assigned to multiple protein groups were removed. Both *Gria1*- and *Gria3* KO datasets revealed similar numbers of peptides and proteins and Coefficient of Variation (CoV) per sample group (Appendix A). In the *Gria1* KO dataset, filtering left 15,954 peptides that mapped to 3051 unique proteins with a CoV of 12.6% and 12.2% in wildtype and *Gria1* KO samples, respectively (Appendix A). In the *Gria3* KO dataset, 15,867 peptides were retained that mapped to 3048 proteins, and revealed a CoV of 12.2% in wildtype and 14.8% *Gria3* KO samples (Appendix A). In the *Gria1* KO dataset, two unique GluA1 peptides were detected, albeit at a 97% lower expression compared to wildtype (Appendix A). Both peptides originated from the N-terminal domain. This is in agreement with a previous report demonstrating low expression of a truncated GluA1 N-terminal fragment in this *Gria1* KO line [41]. *Gria3* KO mice revealed no expression of GluA3 unique peptides.

Differential testing at false discovery rate (FDR)-corrected *p* < 0.05 revealed downregulation of four proteins in the hippocampal proteome of *Gria1* KO mice (Figure 1a). These included known AMPAR subunits and interactors GluA1, GluA2, SynDIG4 and TARP-γ8 (Figure 1a). The proteome of *Gria3* KO mice did not reveal alterations (Figure 1a). Targeted analysis of known AMPAR interactors revealed up-regulation of PSD-95 (or Dlg4), Shisa6 and Shisa7 selectively in *Gria1* KO mice, albeit at low fold-change and statistical significance only without correction for multiple testing (1.09, 1.15 and 1.16; non-FDR *p* < 0.05, respectively) (Figure 1b, Appendix A). Similarly, *Gria3* KO synapses showed selective up-regulation of AP-2 complex subunit mu, MPP2, SynDIG1 and Rap-2b (with fold-changes of 1.09, 1.06, 1.24 and 1.16; non-FDR *p* < 0.05, respectively) (Figure 1b, Appendix A). An additional 19 previously reported interactors revealed no altered expression in either *Gria1*- or *Gria3* KO mice (Appendix A).

Subsequently, selective regulation of GluA1, SynDIG4 and TARP-γ8 in *Gria1* KOs was validated by immunoblotting (Figure 1c). Quantification of Shisa6 revealed a trend of upregulation, without reaching statistical significance (fold-change of 1.22, *p*-value = 0.37) (Figure 1d, Appendix A). Of interest, CNIH-2 was detected with one peptide in wildtypes and *Gria3* KO mice. In *Gria1* KO mice, this peptide failed the quality criteria for quantitative analysis, suggestive of a down-regulation, which was corroborated by immunoblotting (Figure 1c,d, Appendix A). Together, these data revealed a specific subset of AMPA-receptor interactors with robust regulation in *Gria1* KO specifically, suggesting selective binding of these interactors to GluA1-containing receptors.

Next, we performed AMPAR immuno-purifications (IPs) on the *Gria1*- and *Gria3* KO synapse enriched fraction, to assess the interactomes of GluA1/2 and GluA2/3 receptors in a direct manner.

### 3.2. IP-MS of GluA2/3 from Gria1 and Gria3 KO Synapse Extracts Reveal Subunit-Specific Differential Interactors

IP-mass spectrometry (MS) using anti-GluA2/3 in *Gria1* KO mice revealed enrichment of GluA2 and GluA3 (Figure 2a; Appendix A). Strongest co-enrichment was observed for CNIH-2, TARP-γ2 and Noelin1 followed by TARP-γ3, TARP-γ8, FRRS1L, Shisa9, Noelin3, CPT1c and Shisa6 (Figure 2a). In *Gria3* KO mice, IP-MS of anti-GluA2/3 revealed enrichment of GluA1 and GluA2 (Figure 2a). CNIH-2, TARP-γ8 and SynDIG4 were co-enriched with highest abundances, followed by additional interactors FRRS1L, TARP-γ2, Shisa9, Noelin1, Shisa6, CPT1c and TARP-γ3 (Figure 2a).

### 3.3. Validation with Immunoblotting of GluA2/3 IP in the GluA1 Depleted Synapse Extract

To further validate the observations on preferential interactions, independently in wildtype animals, we performed AMPAR IPs on wildtype hippocampus after depletion of GluA1 containing receptors by IP, followed by immunoblotting (Figure 2b). Based on the *Gria1* KO IP-MS data (Figure 2a), removal of GluA1-containing receptors is expected to cause major reduction in levels of SynDIG4 and TARP-γ8, whereas CNIH-2 and TARP-γ2 are expected to be less affected. After protein extraction from a synaptic fraction, GluA1-containing receptors were removed by IP with 33 µg of GluA1 specific antibody in half of the lysates. After antibody incubation, all lysates were incubated two times with 200 µL A/G PLUS agarose beads, and subsequently used for AMPAR-purification with 10 µg anti-GluA2/3 per experiment. Indeed, anti-GluA2/3 revealed a lack of GluA1 immunoreactivity after GluA1-depletion (Figure 2b). In addition, immunoreactivity of SynDIG4 and TARP-γ8 were absent post depletion of GluA1, whereas immunoreactivity remained present for CNIH-2 and TARP-γ2 (Figure 2b). This suggests that SynDIG4 and TARP-γ8 are major interactors of GluA1-containing receptors, in contrast to the remaining GluA3-containing receptors. Taken together, these data demonstrate GluA1/2 containing receptors have a preferred interaction with CNIH-2, TARP-γ8 and SynDIG4 whereas GluA2/3 containing receptors strongly interact with CNIH-2, TARP-γ2 and Noelin1.

### 3.4. Reversed IP Using Anti-SynDIG4 and Anti-TARP-γ8 Antibodies Reveal Their Co-Occurrence with the AMPAR

The AMPAR IP-MS and expression proteomics revealed a strong co-occurrence of TARP-γ8 and SynDIG4 with GluA1/2 (Figure 1). We therefore hypothesized that these AMPAR interactors are part of a shared AMPAR subcomplex. To test this, we first performed IP-MS using anti-TARP-γ2/(4)/8 and anti-SynDIG4 antibodies (Appendix A), with antibody epitope blocking and empty bead controls, and analyzed co-purified known AMPAR interactors [17] (Appendix A). IP-MS using antibodies against TARP-γ2/(4)/8 copurified high amounts of GluA1, GluA2, GluA3, and revealed SynDIG4 as one of the most abundant proteins in TARP-containing complexes (Figure 3a; Appendix A). Additionally, the cornichon proteins (CNIH-2, -3) were enriched with high abundance (Figure 3a). Other interactors were copurified with > 2-fold lower intensity values compared to the AMPAR subunits, SynDIG4 and cornichon proteins (Figure 3a). Anti-SynDIG4 co-purified GluA1/2, CNIH-3, TARP-γ8, TARP-γ2 and Rap-2b (Figure 3a), demonstrating the presence of a TARP-γ8-SynDIG4 assembly by both approaches.

Both TARP-γ8 [48] and SynDIG4 [17] are known to directly bind AMPAR subunits. Similarly, AMPAR interactors FRRS1L and CPT1c bind the AMPAR directly, in addition to binding each other [29]. To test if also TARP-γ8 and SynDIG4 can bind in absence of AMPAR subunits, we purified overexpressed TARP-γ8-myc from HEK293 cells in the presence of SynDIG4-HA (Figure 3b). Indeed, isolation of TARP-γ8-myc revealed co-assembly with SynDIG4 demonstrating these proteins directly interact (Figure 3b).

### 3.5. Combined IP-Blue Native Quantitative Proteomics Demonstrates the Presence of TARP-γ8 and SynDIG4 in an AMPAR Subcomplex

To further scrutinize this TARP-γ8-SynDIG4 assembly as a subcomplex of the AMPAR in the hippocampus, we investigated the migration of TARP-γ8 and SynDIG4 immunopurified native complexes on BN-PAGE followed by mass spectrometry (termed IP-BN-PAGE-MS), as described previously (Appendix A) [44]. Following IP, native complexes were eluted with an epitope-mimicking peptide, mixed with marker proteins and separated by size on a BN-PAGE gel. The gel was cut into consecutive slices that were separately analyzed by mass spectrometry for protein identification and quantification (Appendix A). Protein abundance values were normalized to their max intensity across the gel, and gel slices were numbered relative to the 720 kDa spiked-in marker protein.

In the gel, purified TARP-γ8 and SynDIG4 were expected to co-migrate together with GluA1 in the migration range of the AMPAR at ~720 kDa and higher if they are indeed part of an AMPAR assembly. Figure 3c reveals the GluA1 and GluA2 immunoreactivity of the synaptic extract fractionated on BN gel followed by immunoblotting analysis (Figure 3c). IP-BN-PAGE-MS of anti-TARP-γ2/8 revealed highest abundance of TARP-γ8 between slice −14 till 1, peaking above the 720 kDa spiked in marker protein (slice −3) (Figure 3d). In the same range also SynDIG4 and GluA1 co-migrated, peaking at slightly higher (slice −5) or lower (slice −2) molecular weight, respectively, with large overlapping migration profiles (Figure 3d). Migration of TARP-γ8 bait protein below the 720 kDa marker may result from disassembly in the BN-gel or represent native AMPAR-independent complexes. Similarly, IP-BN-PAGE-MS of SynDIG4 revealed SynDIG4 migration across a broad range of molecular weights above and below 720 kDa; peak abundance above the 720 kDa marker (slice −4), and large overlapping migration profiles of both TARP-γ8 (peaking at slice −6) and GluA1 (peaking at slice −4) (Figure 3e). Taken together, these data are in line with the presence of a TARP-γ8- SynDIG4 containing AMPAR subcomplex.

### 3.6. Super-Resolution Microscopy Shows Co-Localization of TARP-γ8 and SynDIG4 at the Synapse

Super-resolution microscopy was used on hippocampal neurons with anti-TARP-γ8 and anti-SynDIG4 to reveal their colocalization (Figure 4). Approximately 30% of TARP-γ8 immunoreactivity showed immunoreactivity for SynDIG4 (Manders’ coefficient: 0.30 ± 0.017), and 14% of SynDIG4 immunoreactivity showed immunolabeling for TARP-γ8 (0.14 ± 0.009) (Figure 4b). In order to determine if the TARP-γ8 and SynDIG4 colocalization occurs at the synapse, we additionally stained for Homer, a typical post-synaptic scaffold protein and marker of the glutamatergic synapse [49]. A large fraction colocalizing TARP-γ8 and SynDIG4 (59%) overlapped with Homer positive puncta, imaged at confocal resolution (Manders’ coefficient: 0.59 ± 0.02) (Figure 4b,c). Together, these data revealed that these proteins mostly associate at synaptic sites.

## 4. Discussion

In the current study, we analyzed the distinct interactomes of GluA1/2 and GluA2/3 receptors using wildtype and *Gria1* KO or *Gria3* KO hippocampi. Interaction proteomics revealed TARP-γ8, CNIH-2 and SynDIG4 as highest abundant interactors of the GluA1/2 subtype specifically, whereas GluA2/3 IP-MS revealed strongest co-purification of TARP-γ2, CNIH-2 and Noelin1. Further co-expression analysis revealed that TARP-γ8-SynDIG4 directly interact, and STED microscopy showed co-assembly into an AMPAR subcomplex especially at synaptic sites.

In the past decades, multiple AMPAR interactors have been identified [17,32]. Known AMPAR binding partners vary in their interaction strength and stability [17]. The IP-MS protocol used in the current study favored the identification of a subset of established interactors. These included the more stable interacting transmembrane proteins, consistently identified by proteomics studies and which are considered ‘core’ interactors [17,25]. Stabilization of transient interactions by use of a crosslinker before IP-MS could improve coverage of the AMPAR interactome and its analysis in different (KO) conditions in future studies.

Previous IP-MS analysis on the total pool of hippocampal AMPARs revealed TARP-γ8 and CNIH-2 as the most abundant interactors [9]. In the current study, TARP-γ8, CNIH-2 were identified as the highest abundant interactors specifically of the GluA1/2 receptor subtype in addition to SynDIG4. Other interactors, including TARP-γ2 and Noelin1, only revealed a >10 times lower intensity. For the first time, we investigated the interactome of the lower abundant GluA2/3 receptor in isolation. In contrast to the GluA1/2 receptor, GluA2/3 receptors revealed the strongest interaction with TARP-γ2, CNIH-2 and Noelin1. These latter proteins may therefore be of highest interest for functional studies on the GluA2/3 receptor subtype, and GluA3-dependent disease mechanisms, like the induction of Amyloid-β pathology in Alzheimer’s disease models [15]. Previous work revealed the requirement of GluA3-containing AMPAR endocytosis for the synaptotoxic and cognitive effects of Amyloid-β [15]. The exact mechanism underlying the Amyloid-β induced pathway remains unknown, and logically may involve major GluA3 interactors. Interestingly, oligomeric Amyloid-β was shown to be able to bind TARP-γ2 and Noelin1 [50]. As TARP-γ2 and Noelin1 are major interactors of GluA2/3 receptors, these proteins may be interesting candidates for further investigation into the Amyloid-β associated pathway. -Of note, removal of GluA3 in our current study did not affect expression of TARP-γ2, CNIH-2 and Noelin1, suggesting that, conversely, GluA2/3 receptors are not their major interactors.

In the current study, we observed an AMPAR subtype containing both GluA1 and 3 subunits, and potentially GluA2, in the hippocampus. The GluA1/(2)/3 receptor subtype has been observed in previous studies [4,5,17], but is often overlooked. We validated its presence in mouse hippocampus by direct purification with GluA1 or GluA3 specific antibodies. In addition, we revealed SynDIG4 as an interactor of the GluA1/(2)/3 receptor subtype. Whereas GluA1 and SynDIG4 co-purified with GluA3 in wildtype samples, they were both absent in GluA2/3 IP-MS performed on *Gria1* KO mice. Further experiments are necessary to determine additional interactors of the GluA1/(2)/3 receptor subtype.

In addition, we revealed co-assembly of TARP-γ8 and SynDIG4 in an AMPAR subcomplex by TARP-γ8 and SynDIG4 IP-MS and IP-BN-PAGE-MS. IP-MS revealed co-purification of the AMPAR and SynDIG4 when pulling down TARP-γ8. Conversely, pull down of SynDIG4 revealed co-isolation of the AMPAR and TARP-γ8. However, as co-IP on overexpressed TARP-γ8 and SynDIG4 showed that these two proteins can interact directly, the IP-MS experiments do not necessarily demonstrate their co-assembly in an AMPAR subcomplex. To reveal the presence of a TARP-γ8-SynDIG4 containing AMPAR subcomplex, we separated subcomplexes by size using IP-BN-PAGE-MS. TARP-γ8 (~50 kDa) and SynDIG4 (~37 kDa) were expected to comigrate on the gel together with the AMPAR (>720 kDa), at a molecular weight much higher than that of the two proteins combined (87 kDa). Indeed, IP-BN-PAGE-MS revealed comigration of TARP-γ8, SynDIG4 and GluA1 together above 720 kDa.

In a previous study, both TARP-γ8 and SynDIG4 revealed a similar expression profile across biochemical synaptic subfractions, including de-enrichment at the PSD [8]. Our microscopy analysis revealed colocalization of these proteins largely overlapping with Homer positive synaptic puncta, revealing this subcomplex exists mainly at synaptic sites.

The primary function of SynDIG4 is thought be retaining AMPARs extrasynaptically [40]. Upon stimulation, this block may be released, allowing other interactors to transport the receptor into the synapse [40]. In the current study, we identified TARP-γ8-SynDIG4 as part of an AMPAR co-assembly, a direct interaction between TARP-γ8 and SynDIG4 and their colocalization at synaptic sites. Hence, these two proteins may work together in a mechanism of AMPAR release and synapse insertion. Their strong association with the GluA1/2 receptor subtype may underly the typical activity dependent insertion of GluA1/2 receptors into the synapse during fast excitatory transmission [13]. The exact mechanism of AMPAR regulation by TARP-γ8 and SynDIG4, and the interplay between these two proteins, remains to be established. To test the functionality of this AMPAR subcomplex directly, one could block the interaction between TARP-γ8 and SynDIG4 and measure the effects on AMPAR localization with and without stimulation. For this, identification of the TARP-γ8 binding site to SynDIG4 will be necessary, and can be accomplished, for instance, by crosslink mass spectrometry or a peptide array interaction assay [31].

IP-MS of TARP-γ8 revealed a >2.5 times higher abundance ratio between TARP-γ8 (bait) and SynDIG4 (interactor), than observed between SynDIG4 (bait) and TARP-γ8 (interactor) in the SynDIG4 IP-MS experiments. In agreement with this, TARP-γ8 revealed a higher level of colocalization with SynDIG4, than SynDIG4 with TARP-γ8. This indicates that a larger portion of TARP-γ8 protein is associated with AMPAR receptors decorated with SynDIG4, than the other way around. Possibly a small portion of SynDIG4 protein is associated with AMPAR-TARP-γ8, and is additionally part of other AMPAR-(in)dependent interactions.

## Figures and Tables

**Figure 1 cells-11-03648-f001:**
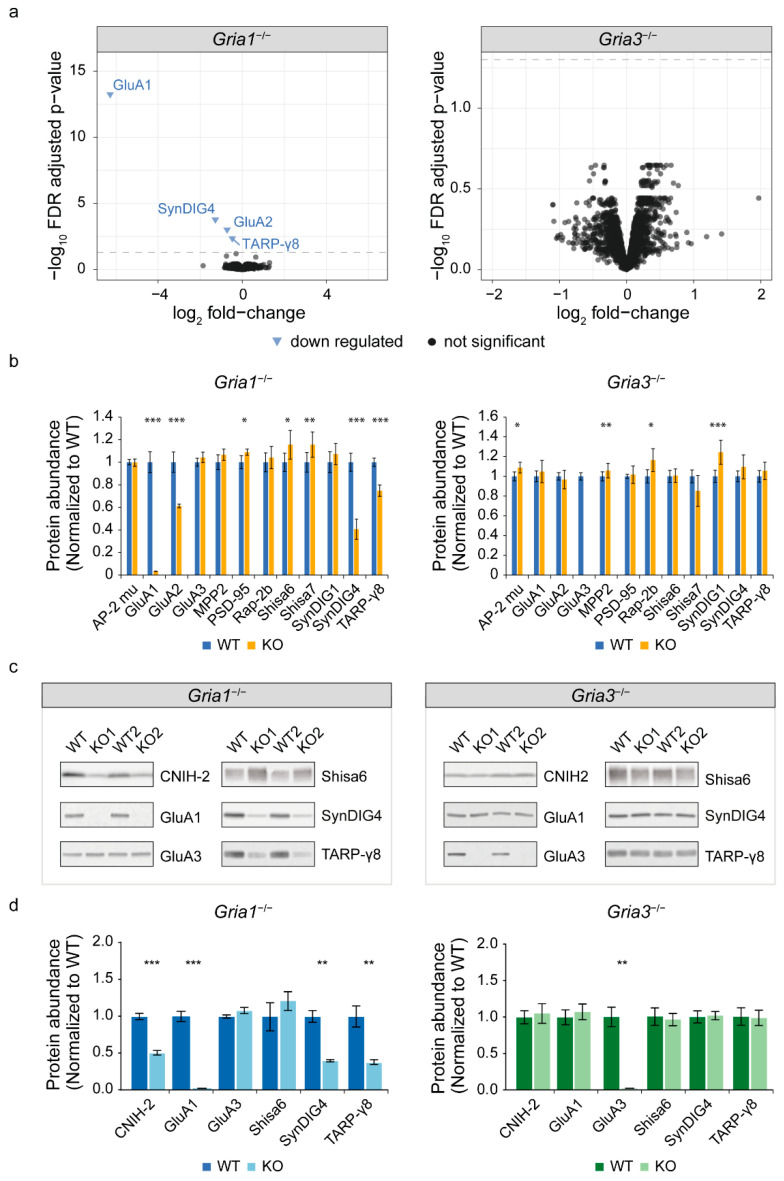
Expression proteomics of *Gria1* and *Gria3* KO synapse enriched fractions. (**a**) Differential abundance analysis revealed 4 proteins with reduced protein expression in *Gria1* KO mice (**left**) (n = 6/condition). No overall changes were observed in *Gria3* KO animals (**right**) (n = 6 WTs, n = 5 KOs). The dotted lines show the eBayes false discovery rate (FDR)-adjusted *p*-value cut-off (5% FDR). (**b**) Selective mass spectrometry data analysis of known AMPA glutamate receptor (AMPAR) interactors revealed specific differential regulation of interactors in *Gria1* KO (n = 6/condition) or *Gria3* KO mice (n = 6 WTs, n = 5 KOs) (eBayes, non-FDR-adjusted *p*-value cut-off < 0.05). AP-2 complex subunit mu is here abbreviated as AP-2 mu. (**c**) Immunoblot validation of mass spectrometry data. (**d**) Quantification of immunoblot validation (n = 6/condition). WT: wildtype; KO: knock-out. Mass spectrometry and immunoblot quantification bar graphs show mean protein abundances normalized to wildtypes ± s.e.m. * *p*-value < 0.05, ** *p*-value < 0.01, *** *p*-value < 0.001, all are non-FDR-adjusted (see Appendix A for details on statistics).

**Figure 2 cells-11-03648-f002:**
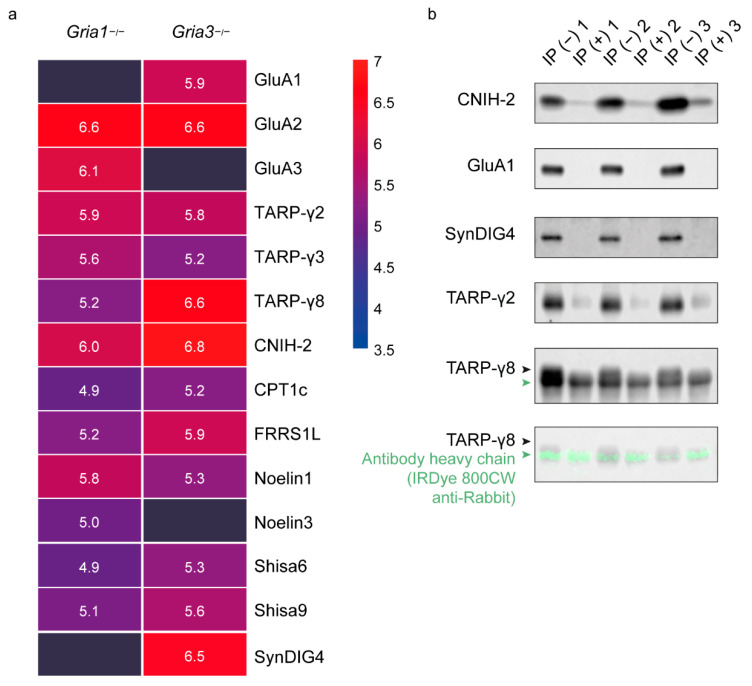
Differential GluA1/2 and GluA2/3 AMPAR interactomes. (**a**) Immuno-purification-mass spectrometry (IP-MS) using anti-GluA2/3 in *Gria1*- and *Gria3* KO hippocampus (n = 3 *Gria1* KOs, and n = 4 *Gria3* KOs). Protein abundances of AMPAR subunits and known interactors are shown as mean log_10_ iBAQ intensity values, and color coded from high abundance (red) to low abundance (blue). (**b**) Validation of *Gria1* KO IP-MS, using anti-GluA2/3 on GluA1-depleted wildtype hippocampus followed by immunoblotting (n = 3/condition). IP− IP without prior depletion of GluA1; IP+: IP with prior depletion of GluA1. The antibody heavy chain of anti-GluA2/3 is visualized in green with IRDye 800CW anti-Rabbit below TARP-γ8, which is visualized with a mouse antibody.

**Figure 3 cells-11-03648-f003:**
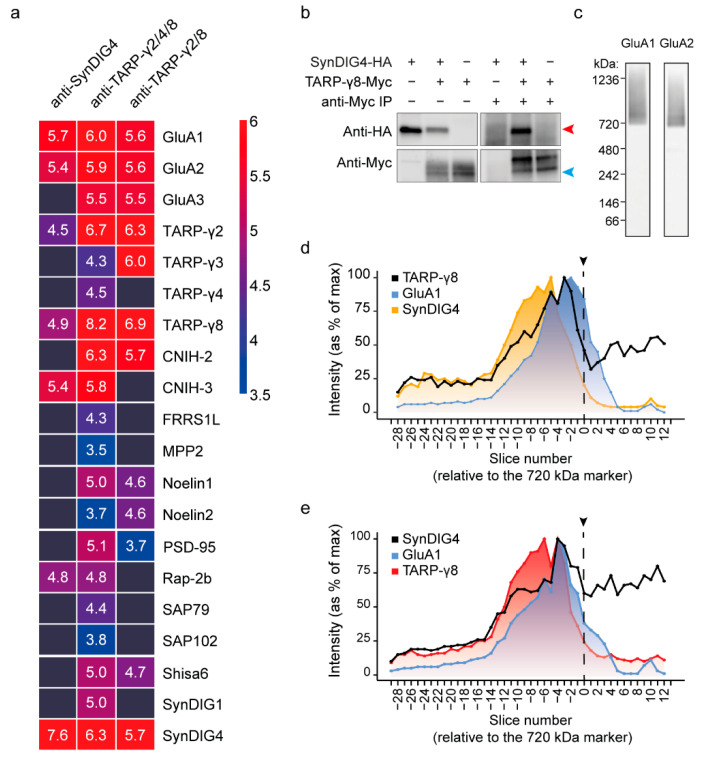
Identification of a TARP-γ8-SynDIG4 containing AMPAR subcomplex. (**a**) IP-MS with antibodies against TARP-γ8 and SynDIG4 in wildtype hippocampus (n = 2 per antibody). Protein abundances of AMPAR subunits and known interactors are shown as mean log_10_ iBAQ intensity values, and color coded from high abundance (red) to low abundance (blue). All values above the 0.75 quantile (>10^6^) were capped to maximum (red) to prevent the bait protein(s) from dominating the scaling. (**b**) TARP-γ8 and SynDIG4 can interact directly without presence of the AMPAR. TARP-γ8-myc (~50 kDa) can directly bind to SynDIG4-HA (~37 kDa) as shown by co-purification from HEK293 cells, using a Myc antibody. Blue arrowhead points to the 50 kDa marker; red arrowhead points to the 37 kDa marker the sizes of TARP-γ8 and SynDIG4, respectively. (**c**) Blue-native PAGE (BN-PAGE)-immunoblot stained with anti-GluA1 and anti-GluA2 reveals separation of native GluA1 and GluA2 containing AMPARs. (**d**) IP-BN-PAGE-MS of anti-TARP-γ8 proteins revealing the migration profile of TARP-γ8, GluA1 and SynDIG4. (**e**) IP-BN-PAGE-MS of anti-SynDIG4 proteins revealing the migration profile of SynDIG4, GluA1 and TARP-γ8. Protein abundance values were normalized to their max intensity across the gel. On the *x*-axis, slices are numbered relative to the 720 kDa spiked-in marker protein The position of the 720 kDa marker is highlighted by the dotted line with arrow.

**Figure 4 cells-11-03648-f004:**
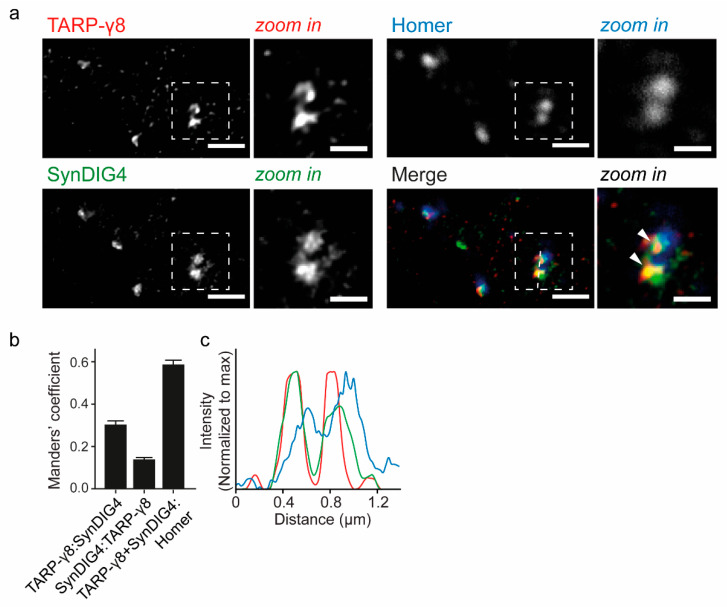
TARP-γ8 and SynDIG4 colocalization analysis on wildtype hippocampal neurons by Stimulated Extinction Depletion (STED) microscopy. (**a**) Dendrites labeled for TARP-γ8 SynDIG4 and Homer with a zoom in on selected puncta (right) (n = 55 fields of view; N = 2 cultures). Arrowheads point out sites of colocalization. (**b**) Manders’ overlap coefficients revealing the fraction of TARP-γ8 positive for SynDIG4; fraction of SynDIG4 positive for TARP-γ8, and the fraction of colocalizing TARP-γ8 and SynDIG4 overlapping with Homer. (**c**) Line graph revealing the relative intensity of TARP-γ8 and SynDIG4 in consecutive TARP-γ8 positive puncta partially overlapping with Homer positive puncta. Mean Manders’ Coefficients are shown ± s.e.m. Image scale bar = 1 μm; Zoom in scale bar = 0.5 μm. Color coding as in (**a**).

## Data Availability

All proteomics data used here have been deposited to the ProteomeXchange Consortium via the PRIDE partner repository with the dataset identifier PXD031603.

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
