# Peer review of "Expression and Interaction Proteomics of GluA1- and GluA3-Subunit-Containing AMPARs Reveal Distinct Protein Composition"

_cells, 2022, doi:10.3390/cells11223648_

Round 1

Reviewer 1 Report

I have reviewed the manuscript entitled “Expression and interaction proteomics of GluA1 and GluA3 subunit containing AMPARs reveal distinct protein composition” by  Sophie J.F. van der Spek and collaborators.

The main point, at least from the title, is that the authors will present new data on proteins that interact with distinct subunits of AMPAR, e.g., GluA1 and GluA3 subunits.

This is a very interesting area of research, and there are a lot of studies being done regarding the identification of AMPARs interacting proteins and their function at the synapse.

In general, the manuscript shows interesting results, very helpful to shed additional light to the knowledge about proteins that specifically interact with distinct AMPAR subunits. This reviewer finds this work appropriate and relevant to the scope of Cells (ISSN 2073-4409). However, various aspects of the manuscript require revision before a final version may be considered for publication. It has several flaws from the abstract to the discussion.

I have several major concerns that need to be addressed.

1)      Abstract.

Although a detailed description of AMPAR subunit composition is provided there is not enough information in regards to their associated proteins. This section finalizes by naming the proteins that interact with GluA1/2 and GluA2/3 subunits with no previous presentation. The authors should provide the reader with information of what are TARP proteins for example… and not just write their abbreviatures.

2)      Introduction.

This section needs improvement.

Abbreviatures must be previously presented.  For example, in line 36 it reads:

“AMPARs have four different subunits GluA1-4 (GRIA1-4), that form distinct 37 combinations.

The reader may not be familiar with what “GRIA 1-4”are. I suggest supporting this sentence with the appropriate information that is missing. For example, by adding “AMPARs have four different subunits GluA1-4 (each encoded by different genes, GRIA1 to GRIA4.GRIA1-4), that form distinct 37 combinations4,5  

Line 50: addà transmembrane AMPA receptor regulatory protein (TARP). Do the same for each protein that is mentioned in the manuscript.

Line 76: Proline-rich transmembrane protein 1 (PRRT1) and in the abstract.

3)      Materials and Methods

Line 88. Please define SNCA. The reader has to guess it is referring to α-synuclein.

Line 156: CO2.. delete extra dot and write CO2.

Unify criteria:

  -sometimes the authors write “minutes” and others min. Lines 101, 141 and 143.

  - sometimes the authors write temperature without a space between grades 37oC and others with a space in between 4 oC.

4)      Results

*Lines 207-213. Which figure shows this result?

*Lines 214-221 describe methodology. It is not a result. Please remove from this section and move it to the materials and methods one.

*Line 237: the reader is addressed to figure 1a. This figure 1 as well as figure 2 are not provided in the manuscript so the reviewer cannot see nor confirm the stated results. Only their legends are provided (lines 258-268, figure 1), (lines 300-306, figure 2)

Please provide these figures!

*Unify criteria: p < 0.05 (line 242)  or p < 0,05 (line 236)

*Line 281: “In parallel to the GluA1-KO IP-MS data, removal of GluA1-containing receptors is expected to cause major reduction in levels of PRRT1 and TARP- γ8, whereas CNIH2 and TARP- γ2 are expected to be less affected.” Please provide a reference for this quote.

*Figure 3 b) Line 328 of the manuscript shows a scheme of an  IP-Blue Native Polyacrylamide Gel Electrophoreses BN-PAGE-MS. This illustration is more useful in the material and methodology section. It could be provided as a supplementary figure.

*Eliminate extra dot in line 346

*Line 375 “Super-resolution microscopy shows the co-localization of PRRT1 and TARP-γ8 with AMPAR near synapse”

Super-resolution microscopy photographs are very small. I suggest converting figures 3 (f-h) to a new and enlarged figure so that the reader can “see” the colocalization of TARP and PRRT1 proteins on hippocampal neurons. A phase contrast image should also be provided.

*Line 382-283 states that “TARP-y8-PRRT1 revealed especially strong signal at the edge of Homer positive puncta (Figure 3h), together revealing these proteins mostly associate near synapses”.

 Previous to this point HOMER is only mentioned in the materials and methodology section without any reference to its biological role or a reference of why the authors choose to analyze colocalization of this postsynaptic density (PSD) scaffold protein.

5)      Discussion.

The authors should add more detail and information in this section. A more profound analysis of the provided results is needed.

Through lines 424-440 the authors propose, based on their results, mechanisms by which the analyzed proteins may associate with each other and regulate AMPAR. A graphical scheme would be helpful to enlighten the reader with the suggested hypothesis.

6)      Supplementary figures

 *Why is the molecular standard visualized in the immunoblots? Shouldn’t we only see the particular protein that is revealed with the appropriate and specific antibody?

* In "Supplemental Figure 4. Antibodies tested for specificity by immunoblot". This figure shows 3 immunoblots (a-c) but figure legends are stated as  (a) , (b) and (d) instead of (c). Please correct! or Is a figure missing? 

Reviewer 2 Report

In their study, “Expression and interaction proteomics of GluA1 and GluA3 2 subunit containing AMPARs reveal distinct protein composition”, the Authors investigated in the hippocampus the interactors belonging to specific AMPAR sub-complexes through specific interactome analysis of the major GluA1/2 and GluA2/3 AMPAR subtypes.  As results, they found that GluA1/2 receptors co-purified TARP-γ8, PRRT1 and CNIH2, while GluA2/3 receptors revealed strongest co-purification of CNIH2, TARP-γ2, and OLFM1. Moreover, the analyses performed suggested that TARP-γ8-PRRT1 interact directly, co-assembling into an AMPAR subcomplex at the synapse.

The manuscript is well written and the results are clearly presented. However, the lacking of a functional assessment of the observed molecular interactions represents a significant limitation of the study. What physiological role do these interactions play in the regulation of hippocampal synaptic transmission and plasticity? Do alterations of these synaptic subcomplexes have a pathogenic role in human pathology?

Considering these issues, in the current form, the study would seem more suitable for a Journal specifically focused on molecular biology.

Round 2

Reviewer 1 Report

The authors have satisfactorily answered to the comments of this reviewer point by point. They have rewritten the abstract and introduction following the reviewer suggestion and the discussion has also been improved considerably.

They incorporated figures 1 and 2 that were missing in the original version. They have enlarged the Super-resolution microscopy photograph as required. All the missing abbreviatures were cited and in addition have been incorporated as a list at the end of the text.

I only found that in figure 4a and in sfigure 5 legends, the reference to Homer (present in the figure) is missing. Please add and amend.

àFigure 4. TARP-γ8 and SynDIG4 colocalization analysis on wildtype hippocampal neurons by STED 380 microscopy. (a) Dendrites labeled for TARP-γ8 and SynDIG4 with a zoom in on selected puncta 381 (right) (n=55 fields of view; N=2 cultures) and SynDIG4 with a zoom in on selected puncta (right) 382 (n=55 fields of view; N=2 cultures). Arrowheads point out sites of colocalization. (b) Manders’ over-383 lap coefficients revealing the fraction of TARP-γ8 positive for SynDIG4; fraction of SynDIG4 posi-384 tive for TARP-γ8, and the fraction of colocalizing TARP-γ8 and SynDIG4 overlapping with Homer. 385 (c) Line graph revealing the relative intensity of TARP-γ8 and SynDIG4 in consecutive TARP-γ8 386 positive puncta partially overlapping with Homer positive puncta. Mean Manders’ Coefficients are 387 shown ± s.e.m. Image scale bar = 1 μm; Zoom in scale bar = 0.5 μm. Color coding as in (a).”

à Supplemental Figure 5. Super-resolution microscopy on hippocampal neurons revealing TARP-γ8 and anti-SynDIG4 colocalization. (a) Dendrites labeled for TARP-γ8 and SynDIG4 with a zoom in on selected puncta (right). Arrowheads point out sites of colocalization. Image scale bar = 1 μm; Zoom in scale bar = 0.5 μm.”

Remove (a) only the description is necessary.

This reviewer finds this work appropriate and relevant to the scope of Cells (ISSN 2073-4409).
